# Structures of the human leading strand Polε–PCNA holoenzyme

Qing He [1,3], Feng Wang [1,3], Nina Y. Yao[2], Michael E. O'Donnell [2] ✉ & Huilin Li [1] ✉

In eukaryotes, the leading strand DNA is synthesized by Polε and the lagging strand by Polδ. These replicative polymerases have higher processivity when paired with the DNA clamp PCNA. While the structure of the yeast Polε catalytic domain has been determined, how Polε interacts with PCNA is unknown in any eukaryote, human or yeast. Here we report two cryo-EM structures of human Polε–PCNA–DNA complex, one in an incoming nucleotide bound state and the other in a nucleotide exchange state. The structures reveal an unexpected three-point interface between the Polε catalytic domain and PCNA, with the conserved PIP (PCNA interacting peptide)-motif, the unique P-domain, and the thumb domain each interacting with a different protomer of the PCNA trimer. We propose that the multi-point interface prevents other PIP-containing factors from recruiting to PCNA while PCNA functions with Polε. Comparison of the two states reveals that the finger domain pivots around the [4Fe-4S] cluster-containing tip of the P-domain to regulate nucleotide exchange and incoming nucleotide binding.

The DNA duplex is a simple elegant structure, yet DNA duplication (replication) is a complicated process in cells, mostly due to the anti-parallel nature of duplex DNA yet the unidirectionality of DNA synthesis[1–3]. Eukaryotes rely on three essential B-family DNA polymerases (Pols) – Polα, Polδ, and Polε – for genome replication[4–6]. These Pols are highly conserved and have distinct functions[4]. Polα is a DNA polymerase/primase complex synthesizing short hybrid RNA/DNA primers to initiate replication[7]. Polα is recruited to the CMG (Cdc45, Mcm2-7, and GINS) helicase either by direct interaction[8] or via a Ctf4 trimer that scaffolds the replisome[9,10]. Polδ is responsible for the synthesis of Okazaki fragments on the lagging strand[11], and Polε is primarily responsible for continuously synthesizing the leading strand[12,13].

As observed for all replicative Pols, Polδ and Polε must partner with a sliding clamp to assemble a "holoenzyme" for processive action[14–20]. Sliding clamps are conserved across evolution[18] and function in many cellular processes beyond DNA replication, including translesion synthesis and DNA repair[21–24]. The eukaryotic sliding clamp is the homotrimeric PCNA (Proliferating Cell Nuclear Antigen). PCNA recruits Pols to DNA primarily via a conserved 8-residue sequence of QxxΨxxθθ (where ψ represents hydrophobicity, θ represents aromaticity, and x represents any residue), which is known as the PCNA Interacting Peptide motif (PIP-motif)[25–28]. Non-canonical PIP motifs have also been identified and adopt divergent sequences[27,28]. PCNA is loaded onto lagging strand DNA by the clamp loader Replication factor C (RFC)[29–31]. Recent cryo-EM studies have revealed how PCNA interacts with yeast and human Polδ to assemble the lagging strand holoenzyme[20,32]. In these structures, Polδ is tethered to one of the three PCNA monomers via the C-terminal PIP motif of the catalytic subunit p125 (Pol3 in yeast), leaving two PCNA monomers unoccupied and available to recruit the flap endonuclease FEN1 or the DNA ligase for Okazaki fragment maturation[32].

Precise details of how PCNA interacts with the leading strand Polε have been unclear, due to a lack of the Polε-PCNA complex structure of any eukaryote, yeast or human. Human Polε is comprised of the largest catalytic subunit POLE1(Pol2 in yeast) and three accessory subunits POLE2-4[33,34] (Fig. 1a). The Pol subunit of eukaryotic Polε appears to have evolved by a gene duplication event, containing a catalytic N-terminal domain (NTD) with both DNA polymerase and 3'-5'

[1]Department of Structural Biology, Van Andel Institute, Grand Rapids, MI, USA. [2]DNA Replication Laboratory and Howard Hughes Medical Institute, The Rockefeller University, New York, NY, USA. [3]These authors contributed equally: Qing He, Feng Wang. ✉e-mail: odonnel@rockefeller.edu; Huilin.Li@vai.org

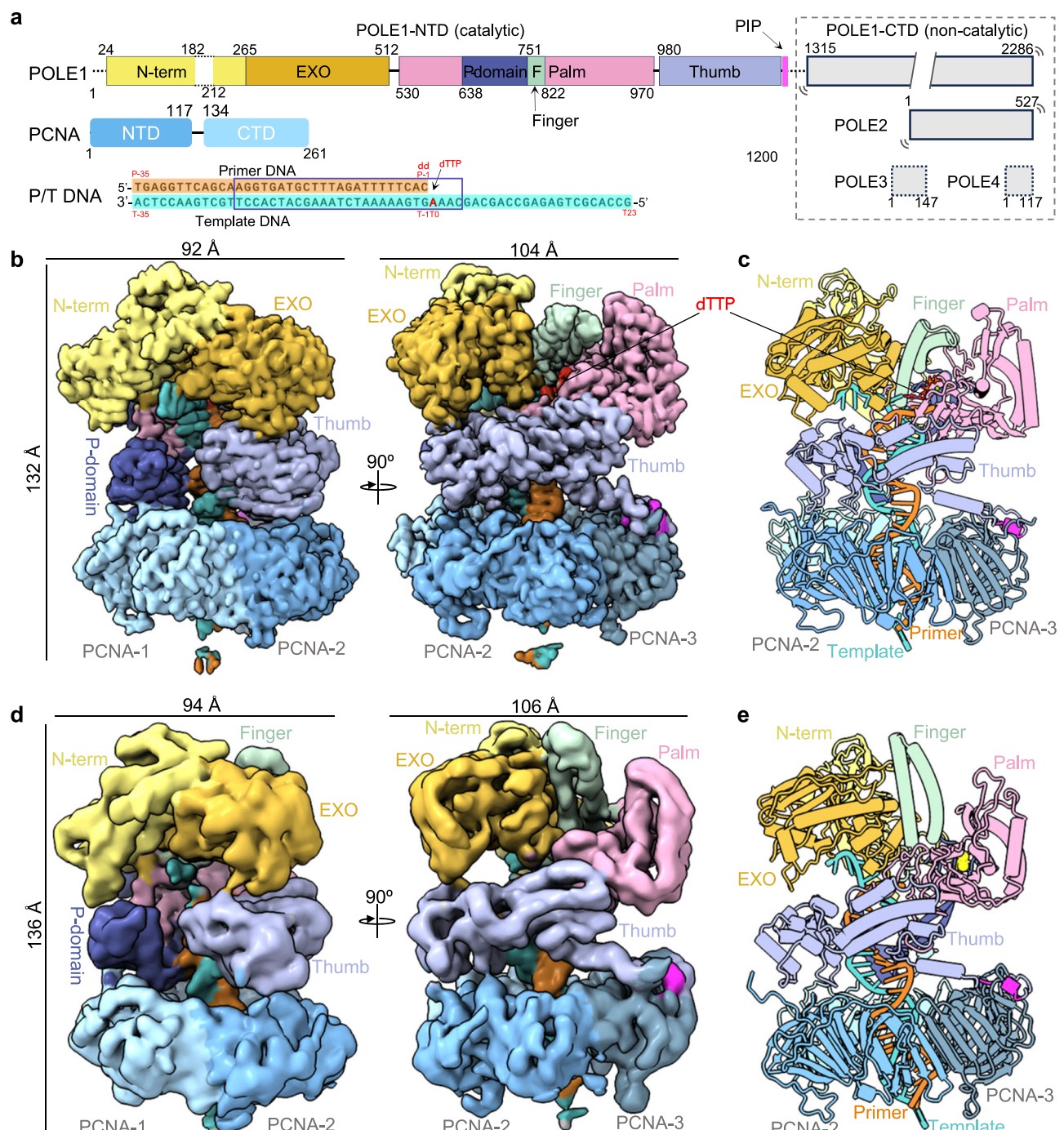

**Fig. 1 | Cryo-EM structures of the human Polε–PCNA–DNA complex. a** Domain architectures of PolE1-4, PCNA, and the nucleotide sequence of the primer/template (P/T) used in this study. Dashed lines indicate disordered regions. The non-catalytic region of POLE1-CTD and POLE2-4 subunits are invisible in all EM maps. The POLE1-NTD is colored by domains, the P/T DNA is colored by strands, and the resolved DNA is boxed. **b**, **c** EM map and atomic model of the Polε–PCNA–DNA complex in the nucleotide bound state. **d**, **e** EM map and atomic model of the Polε–PCNA–DNA complex in the nucleotide exchange state. Individual domains in POLE1-NTD, PCNA protomers, and the incoming dTTP are labeled. The maps and models are colored as in (**a**).

proofreading exonuclease (EXO) activities and a noncatalytic C-terminal domain (CTD) that has homology to DNA polymerases. The POLE1 NTD and CTD are connected by a long linker[34]. POLE2 (Dpb2 in yeast) associates with the POLE1 CTD to form a bi-lobed architecture of Polε[34–36]. POLE3 and POLE4 (Dpb3 and Dpb4 in yeast) are small proteins adopting the histone fold and form a heterodimer[37]. The yeast Dpb3-4 dimer can bridge the two flexibly linked lobes of Polε to form a rigid structure[38]. Both the yeast and human Polε have higher processivity in the presence of PCNA[16,39,40], although how PCNA interacts with Polε has been unknown. POLE1 contains the conserved PIP motif

that presumably binds PCNA, however, the PIP motif is in the middle region of POLE1 between the NTD and CTD—due to the above-mentioned gene duplication (Fig. 1a)—and is different from the PIP motifs in other Pols that are located at the extreme C-termini. Therefore, it would be interesting to more deeply understand how Polε interacts with PCNA.

Unlike Polδ which is recruited to DNA by PCNA, following PCNA loading by RFC[29–31], Polε is recruited to DNA during origin activation before priming, and thus prior to PCNA loading, and its non-catalytic C-domain (CTD) participates in assembly and activation of the

replicative helicase CMG[41–43]. In a leading strand replisome, the Polε CTD associates with CMG, and appears to grant mobility to the Polε catalytic NTD[38,43–45]. The cryo-EM structure of the human Polε catalytic CTD domain and PolE2 bound to CMG have been elucidated[43], while the full-length human Polε structure is currently unknown. Moreover, the crystal structure of the yeast Pol2 catalytic domain[46] as well as the cryo-EM structure of the yeast Polε complex[38] have been reported, revealing a unique P-domain within the Polε catalytic NTD that contributes to the enzyme's processivity[46], similar to the function of PCNA[39,40]. Hogg et al. showed that the P-domain and its interaction with PCNA are both required for enhanced processivity of the yeast Polε[46]. However, how the P-domain interacts with PCNA was unknown as the interaction was not structurally visualized.

Because Polε is recruited to DNA during initiation prior to PCNA loading[47] and thus has proximity to the 3'-OH of the leading strand primer, it is possible Polε binds the 3' end and compromises the ability of RFC to load PCNA onto the initial leading strand 3'-OH. While this is speculative, we note that eukaryotes have evolved the alternative PCNA loader, Ctf18-RFC that binds the Polε catalytic domain and loads PCNA onto the leading strand[39,48–50].

In the current study, we used cryo-EM to determine the human Polε-PCNA holoenzyme structures in the nucleotide bound and nucleotide exchange states. The structures reveal a unique three-point interface between Polε and PCNA, in which all three PCNA subunits are in contact with the Polε catalytic domain, in contrast to the lagging strand Polδ catalytic domain which interacts with only one of the three PCNA subunits. Comparison of the two states shows how the finger domain rotates around the [4Fe-4S]-containing tip of the unique P-domain to regulate the nucleotide exchange and binding events during DNA synthesis. We further demonstrated by mutagenesis and in vitro primer extension that the main interactions between Polε and PCNA are important for activity. Together, this study provides mechanistic insights into the leading strand Polε-PCNA holoenzyme.

## Results

### Cryo-EM of in vitro reconstituted human Polε–PCNA holoenzyme bound to a template/primer DNA

We purified the full-length 4-subunit human Polε complex and the human PCNA separately and reconstituted the Polε–PCNA holoenzyme by mixing them in vitro with an incoming nucleotide dTTP and a DNA substrate containing a dideoxy terminated 35-nt primer annealed to a 59 nt template (Fig. 1a). We introduced two inactivating mutations (D275A and E277A) in the POLE1 EXO catalytic site to prevent degradation of the P/T DNA[51]. Cryo-EM of the in vitro mixture led to two EM maps, a quaternary complex of Polε–PCNA–DNA–dTTP at 2.95 Å and a ternary complex of Polε–PCNA–DNA at 5.01 Å average resolution, respectively (Fig. 1b–d, Supplementary Figs. 1–3). The high-quality map allowed us to build an atomic model for the quaternary complex. The ternary complex model was derived by rigidly-body fitting the quaternary model into the medium-resolution cryo-EM map (Fig. 1c, e, Supplementary Table 1, Supplementary Figs. 2–3). Despite the use of full-length Polε in our in vitro assembly, only the catalytic Polε domain (i.e., POLE1-NTD) is ordered, sitting on top of PCNA with the DNA bound to POLE1-NTD and entering the PCNA ring. But the N-terminal 23 residues and a 29-residue region (aa 183-211) in the POLE1-NTD are not visible, likely due to mobility (Fig. 1a, b). The noncatalytic POLE1-CTD in complex with the other 3 subunits of Polε (i.e., POLE2-4) are also not visible in the EM maps (Fig. 1b–e). This is consistent with several previous observations indicating that the C-terminal half of the catalytic subunit of Polε is flexible relative to the N-terminal catalytic half of the polymerase subunit[34,38,43–45]. We note that successful assembly of the human Polε–PCNA complex requires a nearly stoichiometric presence of the P/T DNA, consistent with the previous report on the yeast Polε[16].

The catalytic POLE1-NTD shares common Pol structural elements such as the finger, palm, thumb, and proofreading exonuclease domains, but also features a unique P-domain connected to the finger domain, as initially observed in yeast Polε[46]. The quaternary human Polε–PCNA–DNA–dTTP structure contains the incoming dTTP that base-pairs with the template in the polymerase site and has the finger domain in the closed conformation. We refer to this structure as the "nucleotide bound state" (Fig. 1b, c). In contrast, the ternary Polε–PCNA–DNA structure lacks the incoming dTTP and the finger domain is in an open position (Fig. 1d, e). We refer to this structure as the "nucleotide exchange state" because the Pol is open and poised to bind a new dNTP. In both states, POLE1-NTD is above the PCNA ring with the duplex DNA emerging from the catalytic core and threading through PCNA (Fig. 1b–e). The nucleotide bound state is 132 Å × 92 Å × 104 Å in size, similar to the 136 Å × 94 Å × 106 Å dimensions of the nucleotide exchange state (Fig. 1b, d).

### Two out of three interactions with PCNA are unique to Polε

A major discovery of our study is the involvement of all three PCNA subunits in interactions with Polε (Fig. 2a–h). This is highly unexpected, as in the previously determined structure of the yeast and human Polδ holoenzymes, the catalytic subunit Pol3 (human p125) interacts with only one of the three PCNA protomers[20,32]. In the nucleotide bound state, the P-domain, the thumb domain, and the C-terminal PIP motif of catalytic POLE1-NTD each interact with a separate PCNA protomer. Quite surprisingly, these three separate interactions are mediated by each of the three PIP-binding pockets of the PCNA homotrimer (Fig. 2a–d), even though the P-domain and thumb domain lacks a PIP motif (Fig. 2a–c). In the case of the P-domain interacting with PCNA-1 (PCNA protomer-1), the ends of two P-domain α-helices insert into the PCNA-1 PIP-binding pocket, with His-684, Arg-685, Gln-689, Tyr-726, and Lys-729 forming five hydrogen bonds (H-bonds) with PCNA-1 Asp-257, Pro-253, His-44, Lys-254, and Asp-41, respectively (Fig. 2b). In the case of the thumb domain interacting with PCNA-2 (PCNA protomer-2), two short thumb loops insert into the PCNA-2 PIP-binding pocket, with thumb residues Glu-1085 and Ser-1117 forming two H-bonds with PCNA-2 Tyr-211 and Ile-255, respectively (Fig. 2c). The interface between the thumb and PCNA-2 is small, and the two PCNA-interacting thumb residues are not well conserved (Supplementary Fig. 4, gray circles). Therefore, the thumb–PCNA interaction is relatively weak, perhaps to ensure the mobility of the thumb domain for catalysis. The five PCNA-interacting P-domain residues are conserved among higher eukaryotes, but three of the five residues are not conserved in yeast (Supplementary Fig. 4, blue triangles), suggesting a weaker interaction in the yeast system.

### Interaction of the PIP motif with PCNA

The POLE1-NTD C-terminus contains the canonical PIP motif plus an extended β-strand (Fig. 2d, e). Both the PIP motif and β-strand engage with PCNA-3. The 8-residue PIP motif is 1180-QKKISELF-1187 and is largely conserved except for the 7th residue Leu-1186 which is an aromatic phenylalanine in the canonical PIP motif (Fig. 2f). The PIP motif of Polε adopts a short $3_{10}$ helix and inserts into the hydrophobic PIP-binding pocket of PCNA-3. Inside the binding pocket, Lys-1181 forms an H-bond with the main-chain oxygen of PCNA-3 Ile-255, Gln-1180 resides in the so-called "Q-pocket" in PCNA-3[52,53], and Ile-1183 and Phe-1187 interact hydrophobically with PCNA-3 (Fig. 2d–g). The extended β-strand following the PIP motif interacts with the PCNA-3 interdomain connecting loop (IDCL) to form a two-stranded β-sheet (Fig. 2d, e). The β-strand is eight residues long (1188-TLEGRRQV-1195) and interacts with the PCNA-3 IDCL β-strand (121-LDVEQLG-127) primarily through backbone hydrogen bonding. However, the two-stranded β-sheet is further stabilized by two H-bonds between Arg-1192 and Glu-1194 of the extended β-strand and Gln-125 and Asp-122 of the PCNA-3 IDCL (Fig. 2e). Such dual-mode interaction has not been

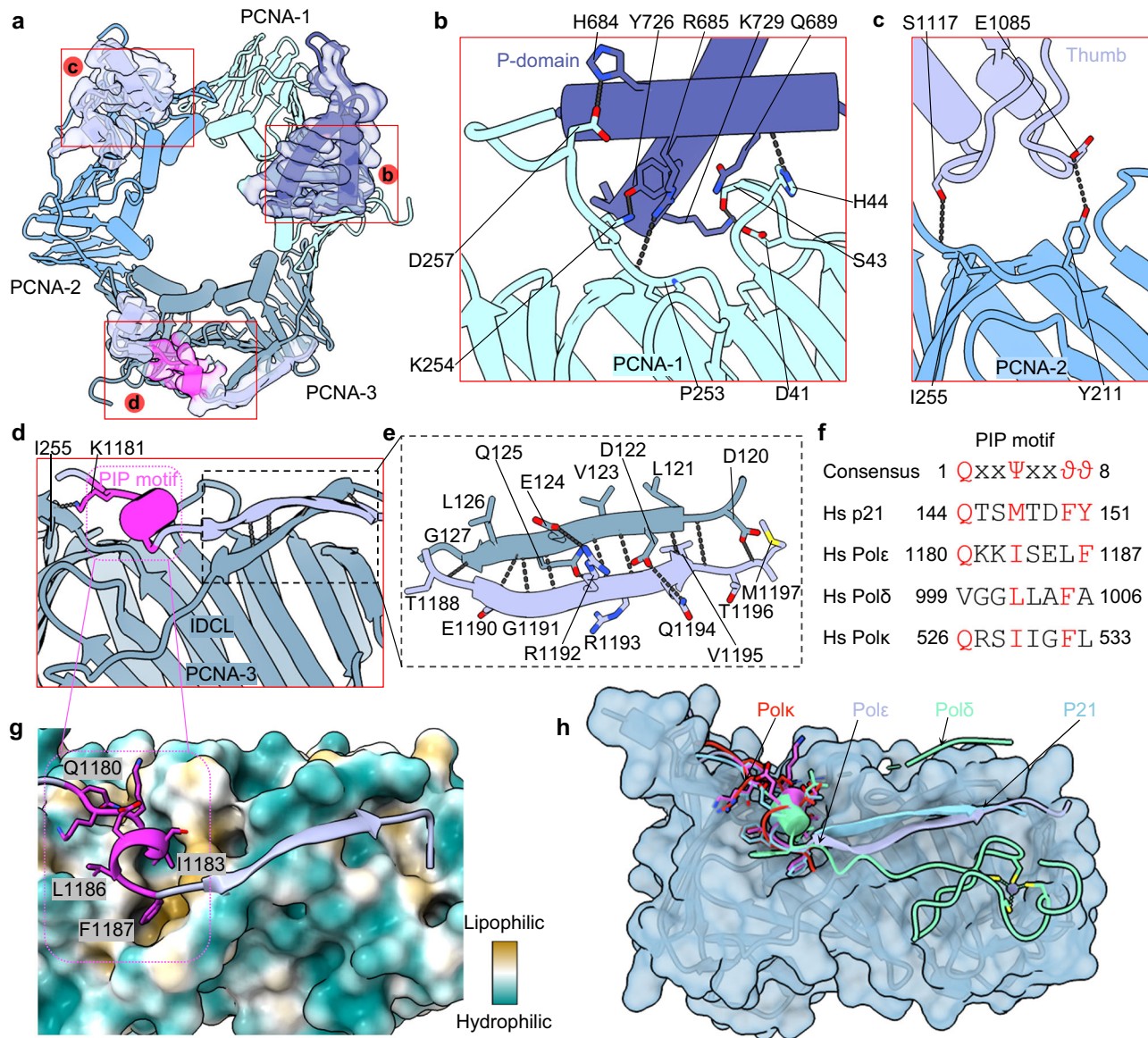

**Fig. 2 | Detailed interactions between the Polε catalytic domain and PCNA.** **a** Cut-open view by removing most of the Polε catalytic domain except for the three binding sites on PCNA, one on each PCNA protomer. **b**–**d** Close-up view of the three interactions between the P domain (**b**), thumb domain (**c**), and C terminus of POLE1 (**d**) with PCNA-1, PCNA-2, and PCNA-3, respectively. **e** Enlarged view of the gray boxed region in (**d**) showing the β-strand interaction between POLE1 and PCNA-3. Key residues are shown as sticks and labeled. Dashed lines mark the H-bonds. **f** Sequence alignments of the PIP motifs of human p21, Polε, Polδ, and Polκ. Conserved residues in the consensus PIP motif are colored in red. **g** Enlarged view of the magenta boxed region in (**d**), showing the interactions of the POLE1 PIP motif with the hydrophobic pocket of PCNA-3 shown in surface view and colored by hydrophobicity. Key PIP motif residues are shown as sticks. **h** Comparison of the PCNA binding modes of the PIP motif of human p21, Polε, Polδ, and Polκ. The extended region of p21, Polδ, and Polκ PIP motifs are shown in aqua, lime, and red, respectively. The extended region of the Polε PIP motif is colored as in (**d**). The PCNA is in cyan.

observed in other Pols but has been observed in the cell cycle regulator p21[54] and in the flap endonuclease FEN1[55]. p21 also contains a canonical PIP motif followed by an extended β-strand that forms a two-stranded β-sheet with the PCNA IDCL[56] (Fig. 2f–h). Polδ contains a noncanonical PIP motif that is extended from one side to the thumb domain and the other side to the C-terminal zinc finger[32]. But the extensions contact only one PCNA subunit and do not form the β-strand interaction with the PCNA IDCL (Fig. 2h). Polκ is a Y-family translesion synthesis polymerase[57]. Polκ has a canonical PIP motif, but the motif is not extended[58] (Fig. 2h).

The interaction between Polε and PCNA is surprisingly unique among all Pol-PCNA complexes characterized thus far, as Polε occupies all three available PIP-binding pockets of a PCNA homotrimer (Supplementary Fig. 5a–d). The total buried interface between Polε

and PCNA is over 2,200 Å², well over twice the interface between Polδ and PCNA (1,022 Å²) or between Polκ and PCNA (733 Å²) (Supplementary Fig. 5d). What is the potential function for such a large interaction? Polδ and Polκ interact with only one PCNA protomer via their respective PIP-motif, leaving the other two PCNA protomers to potentially recruit other PCNA-binding partners. For example, the Polδ–PCNA holoenzyme has two unbound PCNA subunits that may recruit FEN1 or Polκ, even without invoking a conformation change of the polymerization form of Polδ–PCNA–DNA (Supplementary Fig. 5b, c). However, the three-point interaction between Polε and PCNA leaves no space on PCNA for any other PCNA-interacting partners (Supplementary Fig. 5a). It is plausible that Polε has evolved such PCNA contact mode to prevent other PCNA-interacting partners from binding thus interrupting leading strand DNA synthesis in the absence

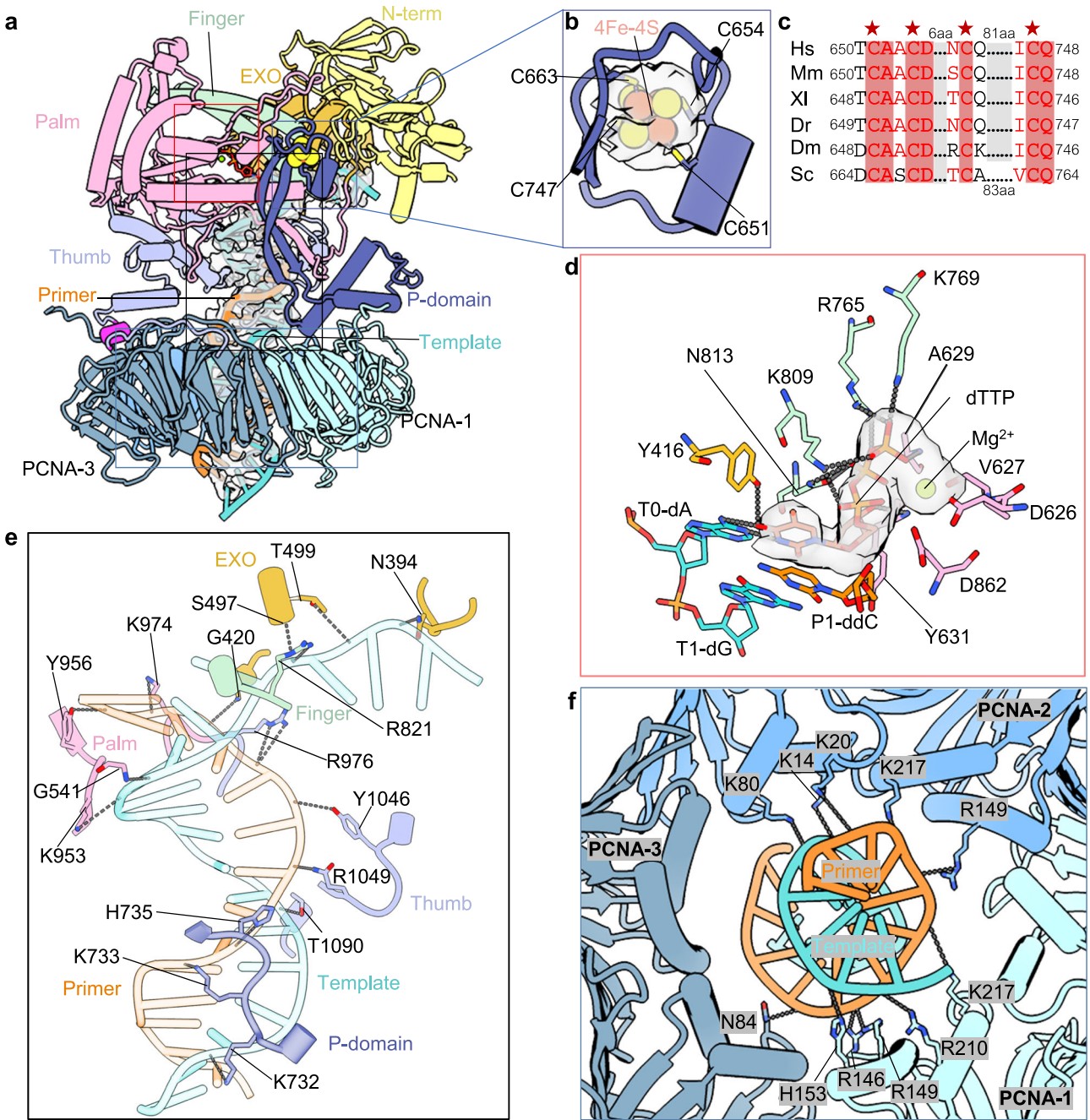

**Fig. 3 | Structural features in the nucleotide bound state of the PolE1-NTD–PCNA–DNA complex. a** Overall structure in cartoon with the DNA density superimposed and shown in transparency. **b** Close-up view of the [4Fe-4S] cluster in the P-domain[59]. The 4Fe-4S cluster is shown in spheres with its EM density in transparent surface. The four coordinating cysteines are in sticks and labeled. **c** Sequence alignment of the [4Fe-4S] coordinating cysteines of human (Hs), mouse (Mm), frog (Xl), fish (Dr), fly (Dm), and yeast (Sc) POLE1. **d** The catalytic site of POLE1. D626, V627, and D862 in the palm, and the three phosphates of dTTP

coordinate a $Mg^{2+}$ ion. The incoming dTTP is coordinated by the $Mg^{2+}$ ion shown in sphere and is base paired with the template dA (T0-dA). Key residues are shown in sticks and labeled. The EM densities of dTTP and $Mg^{2+}$ are shown in transparent surface. **e** POLE1 residues surrounding the DNA are shown as sticks and labeled. Dashed lines mark H-bonds. **f** PCNA-DNA interactions. Key PCNA residues are shown as sticks and labeled. Dash lines mark below 5 Å distances between PCNA and DNA backbone atoms.

of template lesions, thereby improving the processivity of Polε for uninterrupted and continual leading strand synthesis.

## DNA coordination in the nucleotide bound state

In the nucleotide bound state of Polε−PCNA holoenzyme, the DNA is stabilized by both Polε on the top and PCNA on the bottom and has strong EM density (Fig. 3a). Interestingly, the largely exposed middle section of the DNA is supported by the thumb domain from left and the P-domain from right (Fig. 3a). The P-domain contains a 4Fe-4S

cluster at the top region adjacent to the EXO domain and the finger domain[59]. The 4Fe-4S cluster is coordinated by four invariantly conserved cysteines Cys-651, Cys-654, Cys-663, and Cys-747, similar to the coordination in the yeast Polε[46] (Fig. 3a−c). Inside the catalytic pocket, the incoming dTTP has well-defined EM density and forms a base-pair with an adenine of the template strand (Fig. 3d). The dideoxyribonucleotide ddC (P1) at the primer 3′ end stabilizes the incoming dTTP via a π-π interaction between the cytosine and thymine ring (Fig. 3d). The dTTP triphosphate inserts between the palm and

finger to H-bond with the main chain nitrogen atom of Ala-629 of the palm and the finger side chains of Arg-765, Lys-769, Lys-809, and Asn-813. In addition, the dTTP pentose ring forms an H-bond and a π-π interaction with the palm Tyr-631, and the dTTP thymine base forms an H-bond with the Tyr-416 side chain of the EXO domain (Fig. 3d). Only one $Mg^{2+}$ is observed in the Pol active site of the human holoenzyme, and it is coordinated by the dTTP phosphates and the conserved palm residues Asp-626, Val-627, and Asp-862. This coordination is similar to the B-site $Mg^{2+}$ in the crystal structure of the yeast Pol2[46].

Interestingly, the P-domain, thumb, and palm spiral along the minor groove of the duplex DNA, primarily contacting the phosphate backbones. Meanwhile, the finger and EXO domains bind to the extended single-strand region of the template DNA (Fig. 3e). Specifically, the P-domain Lys-732, the thumb Thr-1090, and the palm residues Gly−541and Lys-953 contact the template strand, and the P-domain Lys-733 and His-735, the thumb Arg-976 and Tyr-1046, and the palm Tyr-956 and Lys-974 contact the primer strand. The ssDNA extension of the template is stabilized by the EXO domain Asn-394, Gly-420, Ser-497, and Thr-499, and the finger Arg-821. In the unsharpened cryo-EM map, the template ssDNA density extends to and directly contacts the tip region of the P-domain (Supplementary Fig. 6), indicating that the P-domain plays a role in stabilizing the template strand.

The DNA duplex region is slightly tilted inside the PCNA chamber, resulting in the DNA backbones forming H-bonds or electrostatic interactions (3 Å < distance < 5 Å) with the positively charged Lys-14, Lys-20, Lys-80, Arg-146, Arg-149, His-153, Arg-210, and Lys-217 at the inner surfaces of PCNA-1 and PCNA-2 (Fig. 3f). Of these, the PCNA-1 His-153 and Arg-210 and PCNA-2 Lys-14 and Lys-80 contact the template strand, and the PCNA-1 Arg-146, Arg-149, Lys-217 and the PCNA-2 Lys-20, Arg-149, Lys-217 contact the primer strand, with only the PCNA-3 Asn-84 forming a H-bond with the primer strand (Fig. 3f). These interactions are mostly electrostatic and weak, similar to those observed in the Polκ−PCNA−DNA structure[58], and consistent with the transient nature of the PCNA and DNA interaction.

### Finger domain rotates around the [4Fe-4S] containing tip of the P-domain

The nucleotide exchange state is transient and dynamic, likely accounting for the medium resolution (5.01 Å) of the EM map. But the bound DNA is resolved with clear major and minor grooves features, and the modeled P/T DNA fits the EM density well (Supplementary Fig. 7). In the nucleotide exchange state, the finger domain is in an open position (Fig. 4a), and there is no incoming nucleotide in the Polε activate site (Fig. 4b, Supplementary Fig. 7). Superimposing this state with the above-described nucleotide bound state by the shared PCNA reveals that the finger domain rotates by 25° to close the activate site (Fig. 4a). The finger rotation triggers the N-term and EXO domain to move away by 12.4 Å at the top region. The palm moves toward the finger by 7.0 Å, and the P-domain rotates and moves by 4−5 Å. These protein domain movements lead to the DNA to tilt an additional 5° away from the PCNA axis as compared to the nucleotide exchange state (Fig. 4c).

Because the P-domain is elongated, and its [4Fe-4S] containing upper end is connected to the finger domain, a closer inspection reveals that the [4Fe-4S] containing tip of the P-domain functions as a hinge for finger rotation (Fig. 4d, Supplementary Movie 1). The 25° finger rotation moves the P-domain upper end up by ~4 Å and the P-domain lower end away from the DNA by 3−4 Å. However, the thumb α/β subdomain moves towards DNA, pushing the DNA up ~6 Å at the top and ~3 Å at the bottom (Fig. 4d). The net effect is a stronger interaction between DNA and the Polε active site in the nucleotide bound substrate (Fig. 4a, d). The [4Fe-4S] cluster was previously shown to be essential to the yeast Polε function[59]. This cluster likely plays a structural role to strengthen the hinge to support the finger rotation.

Overall, the transition from the nucleotide exchange to nucleotide bound state moves the DNA down slightly to create an insertion site to accommodate the incoming nucleotide (Supplementary Movie 1). This nucleotide exchange state resembles to the nucleotide pre-insertion state described previously for the bacterial DNA polymerase I[60−62]. Because the incoming dTTP cannot be incorporated into the blocked 3' primer end (ddC), both the nucleotide exchange and nucleotide bound states of the human Polε must chronologically precede the catalysis reaction and belong to the initiation stage of a nucleotide incorporation cycle.

### The main PCNA interaction sites are essential to Polε activity

We next examined the functional importance of the interface between Polε and PCNA. We used POLE1-NTD (aa 1−1200) as the wild-type Polε catalytic domain and designed five mutations as follows (Fig. 5a): Mut-1 (POLE1 aa 1-1188) deletes the extended β-strand that interacts with PCNA-3 IDCL (Fig. 2e), Mut-2 (POLE1 aa 1-1175) deletes both the PIP motif and the extended β-strand (Fig. 2d, e), Mut-3 (aa 1−1200) contains a frequent cancer mutation R685W in the P-domain[63], Mut-4 (aa 1−1200) contains five point mutations (H684A, R685A, Q689A, Y726A, and K729A) in the P-domain that are designed to disrupt the interaction with PCNA-1 (Fig. 2b), and Mut-5 (1−1200 aa) contains four point mutations (Q1180A, I1183A, I1186A, and F1187A) that interrupt the PIP-motif interaction with PCNA-3. Because the thumb domain only weakly interacts with PCNA-2 via two H-bonds, we did not make mutations in this domain. We purified the five mutant proteins individually and compared their primer extension activities with Polε and the PolE1-NTD in the presence or absence of RFC and PCNA (Fig. 5b−d).

The DNA substrate was produced by annealing a 30 nt primer and an 80 nt template (P30/T80) (Fig. 5a). Biotin-streptavidin blocks were added at both ends of the template strand to prevent PCNA from diffusing onto the DNA, where PCNA was loaded by the RFC clamp loader. As expected, Polε alone without PCNA had only weak DNA synthesis activity, and the addition of PCNA and the RFC clamp loader greatly stimulated DNA synthesis by Polε (Fig. 5c, compare lanes 1 and 2). To investigate if the accessory subunits POLE2-4 were required for Polε stimulation by PCNA and RFC we examined the effect of PCNA on DNA synthesis by POLE1-NTD (POLE1 aa 1−1200). We found that the POLE1-NTD activity was slightly lower than that of Polε as noted earlier[46], but the degree of stimulation conferred by PCNA on POLE1-NTD activity was nearly the same as that seen with Polε (Fig. 5c, compare lanes 1−2 with lanes 3−4). This result is consistent with our cryo-EM observation that the regulatory subunits as well as the Pol1-CTD form a flexible lobe that does not interact with PCNA. PCNA stimulated DNA synthesis by the Mut-1 POLE1-NTD lacking the extended β-strand, although the stimulation effect was not as strong as that of the intact POLE1-NTD (Fig. 5c, compare lanes 5 and 6). However, PCNA did not stimulate DNA synthesis by the Mut-2 POLE1-NTD that lacked both the PIP motif and the extended β-strand (Fig. 5c, compare lanes 7 and 8), indicating that the C-terminus of POLE1-NTD is required for PCNA binding. Further, the four-point mutations in the PIP motif (Mut-5) completely abolished PCNA stimulation of DNA synthesis by the Mut-5 POLE1-NTD (Fig. 5c, compare lanes 13 and 14), indicating that the consensus PIP motif is essential to interaction with PCNA.

The P-domain is noted to play a major role in Polε processivity in the absence of PCNA[46], and our structure reveals the domain forms five H-bonds with PCNA, indicating that the P-domain might also confer processivity by PCNA. Mutating all five involved residues in Mut-4 totally disabled POLE1-NTD (Fig. 5c, lanes 11-12), confirming that the P-domain is essential for the polymerase activity of Pol ε. However, a long exposure of the DNA gel shows that the Mut-4 protein produced a small amount of short DNA fragments in the presence of PCNA and RFC, but produced no products in the absence of PCNA (Supplementary Fig. 8). Therefore, PCNA can partially rescue Mut-4 POLE1-NTD, likely due to the presence of other PCNA binding sites. Interestingly,

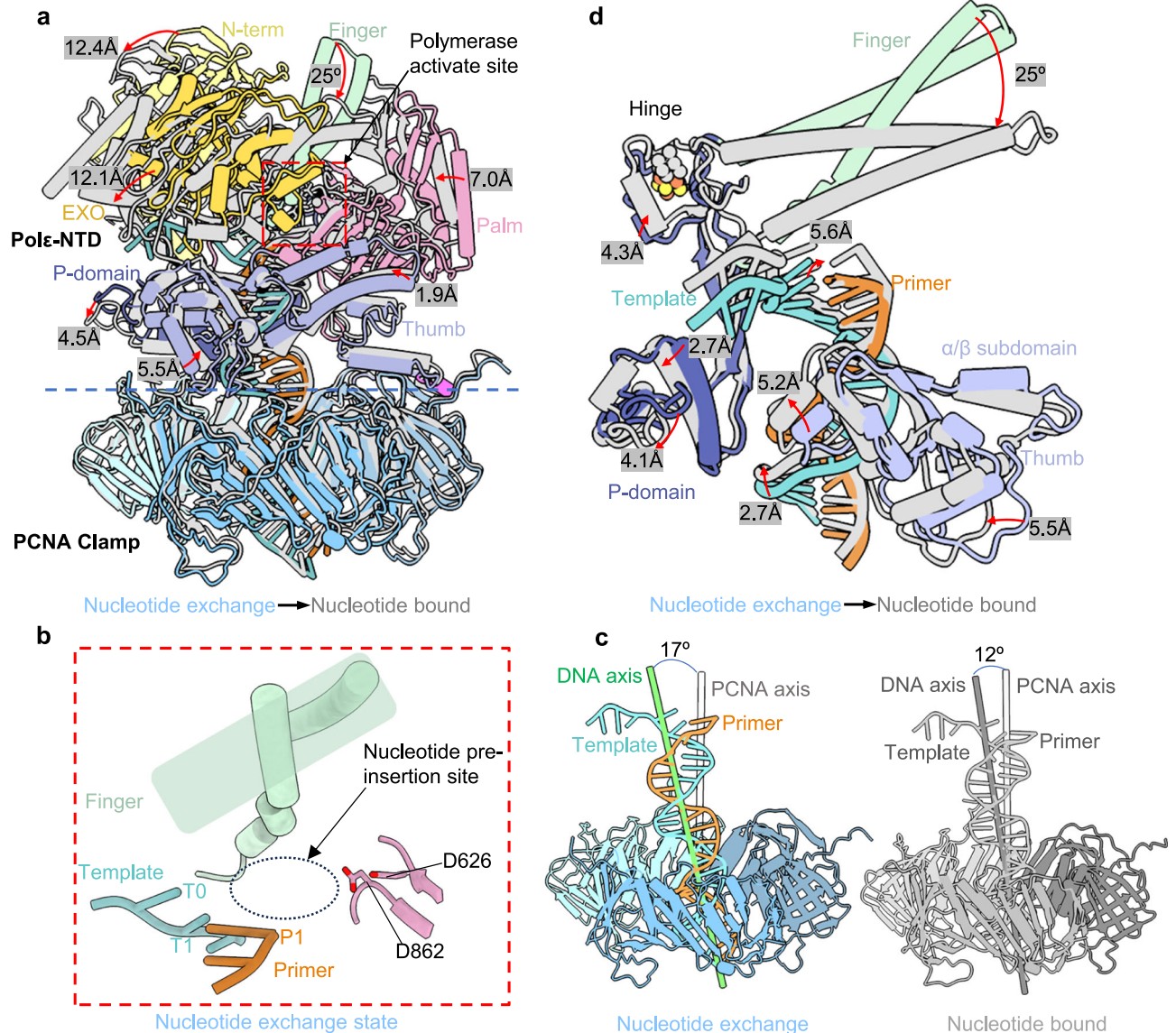

**Fig. 4 | The nucleotide exchange state in comparison with and nucleotide bound state. a** Structures of the nucleotide exchange state (color) and nucleotide bound state (gray) aligned on PCNA. **b** Enlarged view of the polymerase active site region marked by a red square in (**a**). The dashed gray oval indicates the lack of incoming TTP binding in the nucleotide exchange state. **c** DNA orientation with respect to the axis of the PCNA ring in the two states. **d** Close-up view of transition from the nucleotide exchange to nucleotide bound state. The finger rotates by 25 around the [4Fe-4S] rigidified hinge at the tip of the P-domain, causing various movement of the P-domain, thumb, and DNA that lead to incoming dTTP binding. The red arrows in panels (**a**) and (**d**) mark movements between the two states.

one of the five residues (Arg-685) is involved in cancer (R685W)[63]. The Mut-3 protein carrying this single mutation (R685W) had similar DNA synthesis activity as the wild-type POLE1-NTD and was stimulated by PCNA to a similar extent (Fig. 5c, compare lanes 9 and 10). All these PCNA-stimulated polymerase activities are summarized in Fig. 5d. It is clear from this mutagenesis and in vitro activity assay that Polε's interaction with PCNA is dominated by the PIP motif, despite the presence of three different PCNA binding sites in Polε.

## Discussion

Despite the fundamental importance of the human leading strand Polε polymerase, the structure of human Polε, and how Polε from any organism interacts with PCNA to form the functional Polε–PCNA holoenzyme has been missing. This study fills this knowledge gap by determining the structure of the human Polε–PCNA holoenzyme. Our structures have revealed several major biological insights that have important mechanistic implications, including a function for the unique P-domain in binding PCNA and contributing to Polε activity

with PCNA. We also found that Polε binds all 3 protomers of PCNA, which is different from other DNA polymerases.

### How does P-domain contribute to Polε processivity?

Among several reported structures of DNA polymerase holoenzymes[20,32,58], human Polε is unique in having a P-domain that embraces the DNA substrate, and indeed the P-domain was initially demonstrated to encircle the newly synthesized dsDNA and contribute to the enzyme's processivity in the absence of PCNA[46]. In this study, we show that the P-domain not only encircles the duplex region but also contacts the template single-stranded region (Fig. 1, Supplementary Fig. 6). Moreover, the P-domain bottom region directly engages the PCNA clamp (Figs. 2–3). Our primer extension assays of the P-domain mutant protein (Mut-4) confirm the importance of the physical interaction between the P-domain and PCNA for the processivity of human Polε (Fig. 3b, Supplementary Fig. 6), which is consistent with the previous finding in the yeast Polε[46]. Our work affirms the concept that Polε is a PCNA-dependent DNA polymerase. This is not as trivial of a point as

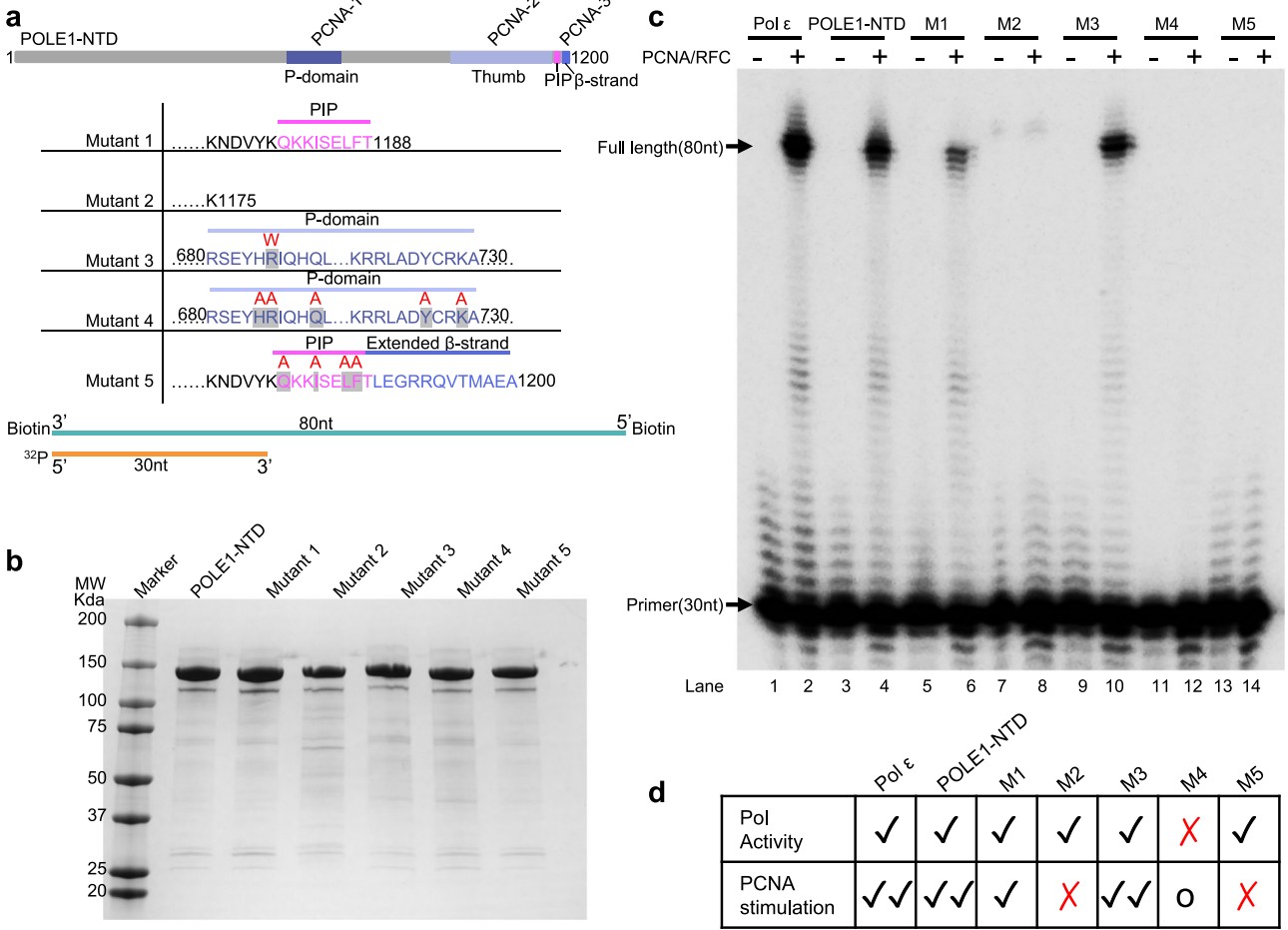

**Fig. 5 | In vitro DNA synthesis by Polε containing five mutations in the catalytic domain.** Schematic view of the five POLE1-NTD mutations (Mut-1 to -5) and the P/T DNA substrate used in the primer extension assays. Mutated residues are shown in magenta above the native sequence. **b** SDS-PAGE gel of the purified wild-type POLE1-NTD and Mut1-5 proteins, three replicate experiments using independent POLE1 samples showed reproducible results. **c** Primer-extension analysis of PCNA-stimulated wild type and mutant Polε on the DNA substrate. Wild-type and mutant forms of Polε were incubated with the substrate in the presence or absence of PCNA at 30 °C for 40 s, and the products were separated on 15% denaturing polyacrylamide gels for 3 h. Experiments were repeated two times. **d** Table summarizing the primer extension assay results. Mut-4 alone has no Pol activity but is slightly stimulated by PCNA.

it may appear, because Polε is physically attached to the CMG helicase via its inactive POLE1-CTD, and it has been unclear if additional attachment via PCNA is needed.

**No room for others**

The homotrimeric PCNA clamp could serve as a tool belt to bind multiple PCNA binding partners at the same time[23,64]. In support of this concept, the archaeal PolD binds two PCNA protomers[19], and Polδ binds one PCNA protomer, leaving room for additional PCNA binding partners[20,32,58] (Supplementary Fig. 5). Indeed, FEN1 can bind to the human Polδ–PCNA for Okazaki fragment maturation[32]. However, the most striking feature in our Polε–PCNA–DNA structure is that the POLE1-NTD interacts with each PCNA monomer, occupying all three PIP binding pockets of PCNA. While our study was conducted independently, we acknowledge the related work by Roske and Yeeles, which also revealed the full occupancy of PCNA by the human POLE1-NTD[65]. This may prevent other PCNA binding proteins from interacting with Polε–PCNA during activity.

POLE1-NTD has the largest interface with PCNA and covers the PCNA ring around the circumference. The tripartite PCNA-binding capability is enabled by the three structural elements: the P-domain, the thumb domain, and the C-terminal PIP motif with an extended β-strand. But the interaction is dominated by the PIP motif and the extended β strand, which bind to PCNA tightly in a manner similar to

p21 and FEN1[56], and the interactions mediated by the P-domain and the thumb are much weaker. So, the three-point interface may not be so much for strengthening the affinity for PCNA and instead, such unique contact may have evolved to physically block the access of other PCNA binding factors to the leading strand. Therefore, the Polε–PCNA appears to have forgone the tool belt mode to synthesize the leading strand DNA in an uninterrupted manner[56].

The DNA is perpendicular to the PCNA plane in the yeast Polδ–PCNA–DNA complex structure, and there is no direct contact between DNA and PCNA inner chamber[20]. However, DNA is somewhat tilted in both nucleotide exchange and nucleotide bound states of the human Polε, resulting in several direct interactions between the DNA and PCNA inner chamber, although these interactions are weak and transient (Fig. 3f). Given the three-point interaction between Pol ε and PCNA, it is likely that the Polε–PCNA complex moves on the DNA in the well-established "sliding" model, similar to the recently described human Polκ sliding on DNA[58].

**Does the catalytic subunit in the Polε–PCNA holoenzyme remain flexible in the leading strand replisome**

The yeast Polε alone in the absence of PCNA exists in two distinct forms: a high population state (95%) in which the catalytic Pol2-NTD is totally flexible, attached to the Pol2-CTD and Dbp2 via a flexible linker region, and a minor "rigid" state in which the linker between Pol2-NTD

and Pol2-CTD forms a "mooring helix" that anchors Dpb3-Dpb4 between the two lobes and thereby stabilizing the complex[38] (Supplementary Fig. 9a). This work raises two fundamental questions: (1) Is the Dpb3-Dpb4 heterodimer associated with Pol2-Dpb2 in the flexible form? and (2) Does Polε, when in the context of a replisome, function in the structured "rigid" form or the flexible form? In this regard, several cryo-EM analyses have shown that the non-catalytic half of Pol2 (POLE1-CTD in humans) and Dpb2 bind to the replicative CMG helicase to assemble the leading strand replisome, while the Pol2-NTD is invisible (i.e., averaged out) and presumably flexibly linked to the CTD[41–44].

In the current human Polε–PCNA–DNA structure, the non-catalytic domain (POLE1-CTD and POLE2-4) is not visible, indicating a flexible form of Polε (Fig. 1a). Superimposition of this structure with the rigid-form of the yeast Polε structure reveals that the thumb and finger domains undergo the most significant changes (Supplementary Fig. 9a-b): the finger domain is closed in the human Polε polymerizing state and is open in the yeast Pol2; the thumb contacts the mooring helix via several H-bonds in the rigid yeast Polε, but in human Polε the thumb domain moves away from the potential mooring helix to bind PCNA (Supplementary Fig. 9b). Therefore, the thumb domain would not be able to interact with the mooring helix in the human Polε–PCNA holoenzyme. Further, the region corresponding to the PCNA IDCL-interacting extended β-strand of the human POLE1-NTD is flexible and links the mooring helix and the PIP motif in the rigid form of yeast Polε (Supplementary Fig. 9c). Therefore, the interaction between the catalytic Polε-NTD and PCNA likely alters the mooring helix, and the destabilized mooring helix renders the catalytic NTD dynamic, in agreement with the observed replisome structures[41–44]. This is consistent with the observation that the rigid yeast Polε structure with a stabilized mooring helix is unable to bind PCNA[16].

## Methods

### DNA constructs and nucleic acids

The cDNAs for the four human POLE subunits were obtained from DNASU Plasmid Repository, and the cDNA encoding human PCNA was obtained from Addgene. The cDNA of POLE1 with an N-terminal 6xHis tag was cloned into the pFL multi-gene expression vector[66]. The cDNAs of POLE2 with an N-terminal 3xFLAG tag, POLE3, and POLE4 were cloned into another pFL multi-gene expression vector following the protocol of biGBac with minor modifications[66]. The POLE1 N-terminal truncation mutants (1-1200aa, 1-1188aa, and 1-1175aa) were tagged with a tandem N-terminal 3 x FLAG-6xHis tag and cloned into the pFast-Bac Dual Expression vector (Thermo Fisher). Other POLE1 mutants were generated by PCR-based mutagenesis. The cDNA of human PCNA was cloned into the pET28a vector with an N-terminal 6xHis tag and a thrombin cleavage site. All constructs were sequenced to ensure no mutations were introduced during PCR and cloning.

The DNA substrates for cryo-EM were chemically synthesized by Eurofins Genomics, which included a 35-nt primer strand (5′-TGAG GTTCAGCAAGGTGATGCTTTAGATTTTTCAC-3′) with a 2′,3′-dideoxycytidine to prevent the 3′ primer extension and a 59-nt template strand (5′-GCCACGCTGAGAGCCAGCAGCAAAGTGAAAAATCTAAAGC ATCACCTTGCTGAACCTCA-3′). An equimolar oligonucleotide of primer and template were mixed to a final concentration of 100 μM in the annealing buffer (20 mM HEPES, pH 7.5, 50 mM NaCl, and 0.5 mM EDTA). The mixture was then subjected to heat denaturation at 95 °C for 10 min, followed by gradual cooling to room temperature. Finally, the P/T DNA was stored at -20 °C before use.

### Protein expression and purification

Polε with a 6xHis tag on POLE1 N-terminus and a 3 x FLAG tag on POLE2 N-terminus and all POLE1 mutants with an N-terminal tandem 3xFLAG-6xHis tag were expressed using the Bac-to-Bac Baculovirus expression system (Thermo Fisher). For Polε expression, the baculoviruses encoding POLE1 and the POLE2-4 subcomplex were co-infected with the Gibco™ Sf9 cells (cat. no. 11496015) for 65-72 h at 27 °C with constant shaking at 115 rpm. To purify the Polε complex, harvested insect cells were lysed by sonication in lysis buffer (25 mM HEPES, PH7.5, 250 mM NaCl, 1 mM Mg-Acetate (MgAc), 5% Glycerol, 1 tablet EDTA-free protease inhibitors cocktail), centrifuged in a Ti-45 rotor at 125,440 x g for 1 h, and the supernatant was incubated with 0.8 mL FLAG-antibody-coated beads at 4 °C for 2–3 h. Beads were washed twice with 50 mL lysis buffer, and the proteins were eluted with 8 mL of lysis buffer plus 0.2 mg/mL FLAG peptides. The eluted proteins were concentrated using centrifugal concentrators (Amicon, 100 kDa) and further purified by size exclusion chromatography (SEC, Superose 6 Increase, GE Healthcare) in buffer containing 25 mM HEPES, PH 7.5, 200 mM NaCl, 1 mM MgAc, and 1 mM DTT. The purified Polε was concentrated to 3.7 mg/ml and stored at -80 °C.

For POLE1 mutants, the insect cells were lysed and centrifugated as Polε, then the supernatant was collected and loaded into a 5 ml Ni-NTA column (Cytiva). The proteins were eluted using lysis buffer plus 300 mM imidazole. All mutants were further purified by size exclusion chromatography through a Superdex 200 column (GE Healthcare) in buffer 25 mM HEPES, PH 7.5, 200 mM NaCl, 1 mM MgAc, and 1 mM DTT. The human PCNA (hPCNA) was expressed in *E. coil* BL21 as described previously[50], hPCNA was induced by 0.3 mM IPTG at 16 °C for 18 h. The harvest cells in the lysis buffer (25 mM HEPES, PH 7.5, 200 mM NaCl, 1 mM DTT, and 5% Glycerol) were lysed by a homogenizer (SPX Corporation) and was centrifugated at 34,572 x g at 4 °C for 1 h, then his-tagged hPCNA was purified through Ni-NTA column (Cytiva) like the POLE1 mutants. Finally, hPCNA was subjected to a Superdex 200 column (GE Healthcare) in buffer 25 mM HEPES, PH 7.5, 200 mM NaCl, and 1 mM DTT. The purified hPCNA was concentrated to 3.0 mg/ml and stored at -80 °C.

### Primer extension assays

Reactions utilized an 80mer template that contained biotin at each end to block PCNA sliding. The template was primed with a $^{32}$P-30mer primer annealed to the extreme 3′ end of the template strand. The DNA was then blocked with a 20-fold excess of streptavidin tetramer. This DNA substrate (10 nM) was incubated with 1 nM Polε, 100 nM PCNA, and 5 nM RFC in buffer containing 40 mM Tris-Cl (pH 7.5), 8 mM MgCl$_2$, 10% glycerol, 100 μg/ml BSA, 1 mM DTT, 130 mM NaCl, and 1 mM ATP at 30 °C for 5 min. Then DNA synthesis was initiated upon adding each dNTP at 100 μM. After 40 s, each reaction was quenched with 0.5% SDS, 20 mM EDTA, and analyzed on a 15% urea-denaturing PAGE sequencing gel.

### Cryo-EM grid preparation and data collection

To assemble the human Polε–PCNA–DNA complex in vitro, 1 μM Polε was combined with 3 μM PCNA and 1.2 μM P/T DNA substrates in the presence of 0.5 mM dTTPs at room temperature for 10 min, and then the mixture was incubated on ice for 2–4 h before preparing the cryo-EM grids. Increasing DNA ratio by using 10 μM DNA in the reconstitution significantly reduced the amount of the assembled human Polε–PCNA–DNA complex. To mitigate the preferred particle orientation issue, 0.02% octyl β-d-glucoside was added into the mixture right before preparing cryo-EM grid. Holey carbon grids (Quantifoil Au R1.2/1.3, 400 gold mesh) were glow-discharged in the Ar/O$_2$ mixture for 30 s using a Gatan 950 Solarus plasma cleaning system, and 3ul droplets of the reaction mixture including 0.02% octyl β-d-glucoside were applied to the grids. Sample vitrification was conducted with the following blot settings: blot time of 5 s, blot force of 3, wait time of 5 s, sample chamber temperature of 8 °C, and chamber humidity of 100%, using the Vitrobot Mark IV system (Thermo Fisher Scientific). The EM grids were flash-frozen in liquid ethane cooled by liquid nitrogen. Cryo-EM data were collected on a 300 kV Titian Krios electron microscope controlled by SerialEM in multi-hole mode. A 20° tilted cryo-EM dataset was also collected to further alleviate the preferred

particle orientation problem. Images were captured at a microscope magnification of 105,000 × and an objective lens defocus range of −1.2 to −1.6 μm, and the micrographs were recorded in a K3 direct electron detector (Gatan) operated in the super-resolution video mode, corresponding to a super-resolution pixel size of 0.414 Å. During a 1.0 s exposure time, a total of 50 frames were recorded with a total dose of 60 e-/Å$^2$.

## Image processing and 3D reconstruction

A total of 17,175 untilted and 17,773 tilted raw movie micrographs were collected and motion-corrected using the program MotionCorr 2.0 with 2 × binning, resulting in a pixel size of 0.828 Å/pixel. The motion-corrected micrographs were imported into cryoSPARC (version 4.2.1) to perform the patch-based contrast transfer function (CTF) estimation. A total of 13,114 micrographs and 13,936 tilted micrographs with CTF signal extending to 4.0 Å were retained for further processing (Supplementary Fig. 1d). Blob-based auto-picking (100–160 Å diameter) in cryoSPARC was implemented to select initial particle images and generate 2D templates for subsequent template-based particle picking. A total of 15,363,577 raw particles were automatically picked and 4 x binned extracted. Several rounds of 2D classifications were then performed and particles in the classes with clear structural features were selected. 994,650 raw particles extracted with a box size of 360 pixels were used to calculate five starting 3D models. Two bad 3D reconstructions and one low-resolution class were discarded as junk particles. Two 3D reconstructions with best resolutions were chosen to perform 3D refinement, separately. Particles in the first 3D class were used to perform a new round of 3D classification, one of the five subclasses without clear 3D features was discarded. The other four classes were selected to perform the homogeneous and non-uniform refinements resulting in the final 3D map with a clear closed finger density at an overall resolution of 2.95 Å. The second 3D class had clear open finger and DNA densities, and this class was subjected to a new round of 2D classification to remove bad particles, the selected particles were used for homogeneous and non-uniform 3D refinements that resulted in the final 3D map at an overall resolution of 5.01 Å.

## Model building, refinement, and validation

Crystal structure of the yeast Pol2 (PDB ID 4M8O) was used as a starting model to predict the structure of human POLE1-NTD with a closed finger domain by the Robetta server[67]. The human POLE1 with an opened finger domain was predicted by AlphaFold2[68] and used to build the model of the nucleotide exchange state of the Polε–PCNA–DNA complex. The predicted structures of POLE1-NTD and human PCNA were manually docked into the EM maps in UCSF ChimeraX[69]. The models were manually adjusted and rebuilt to fit the EM map in Coot[70]. The P/T DNA model was built in coot[70], and the EM density features in the nucleotide bound state allowed unambiguous assignment of the DNA sequence at the 3'-P/T DNA junction inside the active site of POLE1-NTD. The refined P/T DNA model in the nucleotide bound state was rigidly fitted into the 3D map of the nucleotide exchange state, and the DNA model fitted well with the EM density. The unresolved DNA region was omitted in both states. The built models were subjected to several iterations of real-space refinement in PHENIX[71] and manual adjustment in Coot[70]. Finally, all atomic models were validated using MolProbity[72]. The EM maps were sharpened by deepEMhancer[73] for figure presentation. The 3D reconstruction and model refinement statistics are listed in Supplementary Table 1. Structural figures were prepared in the UCSF ChimeraX[69].

## Reporting summary

Further information on research design is available in the Nature Portfolio Reporting Summary linked to this article.

## Data availability

The two cryo-EM 3D maps of the Polε–PCNA–DNA complex generated in this study have been deposited in the EMDB under accession codes EMD-44358 (the nucleotide bound state, 2.95 Å) and EMD-44357 (the nucleotide exchange state, 5.01 Å). Their corresponding atomic models have been deposited in the Protein Data Bank under accession codes 9B8T and 9B8S. Previously published structures used in model building and structural comparison are available in PDB with accession codes: 4M8O (the catalytic core of yeast Pol2), 6TNY (the human Polδ–PCNA–DNA complex), 7NV0 (the human Polκ–PCNA–DNA complex), and 6WJV (the rigid yeast Polε holoenzyme structure). Source data are provided with this paper.

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

## Acknowledgements

Cryo-EM data were collected at the David Van Andel Advanced Cryo-Electron Microscopy Suite at the Van Andel Institute. We thank G. Zhao and X. Meng for their help with data collection. This study was supported by the US National Institutes of Health grants GM148159 (to M.E.O) and GM131754 (to H.L.), Howard Hughes Medical Institute (to M.E.O), and Van Andel Institute (H.L.).

## Author contributions

M.E.O. and H.L. conceived the project. Q.H. and F.W. prepared the protein samples and performed EM experiments and data analysis. Q.H. and F.W. built the atomic models. N.Y.Y. carried out the in vitro activity assays. Q.H., F.W., M.E.O., and H.L. analyzed the data and wrote the manuscript.

## Competing interests

The authors declare no competing interests.
