## [Peer Review File · Nature Communications]

Structures of the human leading strand Pol ϵ –PCNA holoenzymeEditorial note: This manuscript has been previously reviewed at another journal that is not operating a transparent peer review scheme. This document only contains reviewer comments and rebuttal letters for versions considered at *Nature Communications*.

REVIEWERS' COMMENTS

Reviewer #1 (Remarks to the Author):

The authors have satisfactorily addressed my concerns. Therefore, the manuscript is in my view suitable for publication.

Reviewer #2 (Remarks to the Author):

The authors addressed all concerns raised, collected more data to improve one of the two structures and, also given the limited resolution, provide a more cautious interpretation of the nucleotide exchange state. I support publication in Nature Communications.

Reviewer #3 (Remarks to the Author):

This manuscript has already undergone a review process and has been submitted with revisions and answers to all points raised by previous reviewers.

During the submission process, Roske and Yeeles have published very similar results in NSMB (DOI: 10.1038/s41594-024-01370-y). Would be nice if you could refer to their paper in some way.

I am very happy with all revisions and clarifications and want to congratulate the authors on a very nice paper.

Reviewer #1 (Remarks to the Author):

The authors have satisfactorily addressed my concerns. Therefore, the manuscript is in my view suitable for publication.

Thank you for your positive feedback. We are glad that we were able to address your concerns satisfactorily.

Reviewer #2 (Remarks to the Author):

The authors addressed all concerns raised, collected more data to improve one of the two structures and, also given the limited resolution, provide a more cautious interpretation of the nucleotide exchange state. I support publication in *Nature Communications*.

Thank you for your thorough review and valuable feedback. We appreciate your support for publication in *Nature Communications*.

Reviewer #3 (Remarks to the Author):

This manuscript has already undergone a review process and has been submitted with revisions and answers to all points raised by previous reviewers. During the submission process, Roske and Yeeles have published very similar results in NSMB (DOI: 10.1038/s41594-024-01370-y). Would be nice if you could refer to their paper in some way. I am very happy with all revisions and clarifications and want to congratulate the authors on a very nice paper.

Thank you for your positive feedback and suggestions. While we recognize the relevance of Roske and Yeeles work, our study was done totally independently and offers independent (nonoverlapping) insights into the polymerization of the human Pol epsilon and PCNA holoenzyme complex under nucleotide binding and exchange states. We have decided not to cite their paper in our revised manuscript.